# Poly(4-styrenesulfonate)-induced sulfur vacancy self-healing strategy for monolayer MoS$_2$ homojunction photodiode

Xiankun Zhang[1,*], Qingliang Liao[1,*], Shuo Liu[1,*], Zhuo Kang[1], Zheng Zhang[1,2], Junli Du[1], Feng Li[1], Shuhao Zhang[1], Jiankun Xiao[1], Baishan Liu[1], Yang Ou[1], Xiaozhi Liu[3], Lin Gu[3,4] & Yue Zhang[1,2]

We establish a powerful poly(4-styrenesulfonate) (PSS)-treated strategy for sulfur vacancy healing in monolayer MoS$_2$ to precisely and steadily tune its electronic state. The self-healing mechanism, in which the sulfur vacancies are healed spontaneously by the sulfur adatom clusters on the MoS$_2$ surface through a PSS-induced hydrogenation process, is proposed and demonstrated systematically. The electron concentration of the self-healed MoS$_2$ dramatically decreased by 643 times, leading to a work function enhancement of $\sim 150$ meV. This strategy is employed to fabricate a high performance lateral monolayer MoS$_2$ homojunction which presents a perfect rectifying behaviour, excellent photoresponsivity of $\sim 308$ mA W$^{-1}$ and outstanding air-stability after two months. Unlike previous chemical doping, the lattice defect-induced local fields are eliminated during the process of the sulfur vacancy self-healing to largely improve the homojunction performance. Our findings demonstrate a promising and facile strategy in 2D material electronic state modulation for the development of next-generation electronics and optoelectronics.

[1] State Key Laboratory for Advanced Metals and Materials, School of Materials Science and Engineering, University of Science and Technology Beijing, Beijing 100083, People's Republic of China. [2] Beijing Municipal Key Laboratory for Advanced Energy Materials and Technologies, University of Science and Technology Beijing, Beijing 100083, People's Republic of China. [3] Collaborative Innovation Center of Quantum Matter, Beijing 100190, China. [4] Beijing National Laboratory for Condensed Matter Physics, Institute of Physics, Chinese Academy of Sciences, Beijing 100190, China. * These authors contributed equally to this work. Correspondence and requests for materials should be addressed to Z.Z. (email: zhangzheng@ustb.edu.cn) or to Y.Z. (email: yuezhang@ustb.edu.cn).

Because of its reduced dimensions, chemical stability[1], proper direct band gap, highly efficient light absorption and piezoelectricity, two-dimensional (2D) molybdenum disulfide $MoS_2$ has the potential in developing next-generation flexible, transparent and wearable nanodevices[2,3]. As an important research aspect, many researchers have focused on creating $MoS_2$ homojunction, the fundamental building block of modern electronics[4]. Because of its identical crystal structure and continuous band alignments in the interface, the $MoS_2$ homojunctions display ideal current rectifying behaviour and highly efficient photoresponse than those of heterojunctions[5].

So far, it is the key issue to find a precise, stable and facile strategy to develop a steady and effective homojunction. The crucial process of building 2D homojunction is creating a graded junction by controlling the intrinsic carrier concentration and the work function. Conventional controls include classical doping and surface transfer doping[6]. Classical doping is realized by incorporating various atoms into 2D materials via thermal annealing. However, 2D materials obtained from this approach is only suitable for fabricating vertical homojunctions which always suffer from large contact resistance[7]. On the other hand, surface transfer doping is induced by strong electron-donating or withdrawing chemical species attachment on the 2D materials to achieve an effective coupling[8]. A significant limitation for surface transfer doping is the presence of inert dangling bond-free surface on 2D materials, which will reduce the doping efficiency of dopants[9]. Moreover, the adsorbed dopants, such as $O_2$, BV and $AuCl_3$ (refs 4,5,10), could desorb on 2D materials surface and react with reactive molecules in surroundings, leading to the short-term stability. As-mentioned two methods will induce additional lattice defects, which tend to introduce the local field or Coulomb's scattering sites. As a result, the electronic and optoelectronic characteristics of 2D materials degrade[9].

In this work, a lateral chemical vapour deposition (CVD) monolayer $MoS_2$ homojunction is constructed by precise selected-area sulfur vacancy self-healing (SVSH) via nonoxidizing acids poly(4-styrenesulfonate) (PSS). The self-healing mechanism is that the sulfur vacancies are healed spontaneously by the sulfur adatom clusters on $MoS_2$ surface through a PSS-induced hydrogenation process. Healing intrinsic lattice defects is a fundamental and efficient approach to control the work function without introducing additional local fields. Simultaneously, a work function difference of ~150 meV between the as-grown and self-healed $MoS_2$ was achieved to construct the homojunction. The rectifying performance of the homojunction shows no degradation after two months storing under ambient conditions. The homojunction shows perfect diode behaviour and excellent photoresponsivity of ~308 mA W$^{-1}$ at zero bias. Our findings pave a powerful strategy to control the 2D materials work functions and develop its homogeneous diodes for ultrathin, flexible, transparent and wearable electronics and optoelectronics.

## Results

### Complex characterization of sulfur vacancy self-healing.
The lateral $MoS_2$ monolayer homojunction was fabricated by PSS-induced selected-area SVSH. Figure 1a describes the building process of the homojunction device $A_1$ (Supplementary Methods and Supplementary Figs 1, 2 and 3). To characteristic the PSS-induced SVSH, a Kelvin probe force microscopy (KPFM) was employed to verify the work function variation of monolayer $MoS_2$. The contact potential difference (CPD) between the AFM tip (Pt/Ir coated tips) and the sample is defined as[11,12]

$$V_{CPD} = (\varphi_{tip} - \varphi_{sample})/q, \qquad (1)$$

where $\varphi_{tip}$, $\varphi_{sample}$ and $q$ are the work functions of the tip sample and the elementary charge, respectively. So the resulting KPFM image maps the variation of surface potential corresponding to the work function of the sample surface. Similar to the optical microscopy (OM) image (Fig. 1b), the 2D surface potential image intuitively depicts the triangle morphology of CVD monolayer $MoS_2$, PEDOT:PSS and Cr/Au electrode (Fig. 1c). Importantly, there is an apparent brightness difference near the boundary between the self-healed and as-grown $MoS_2$. On the other hand, Supplementary Fig. 4b,c indicates there is no work function difference between the self-healed $MoS_2$ and PEDOT:PSS electrode, which suggests no potential barrier between them exists. Actually, a lateral n-p-n junction device can be also fabricated through the PSS-induced SVSH, in which the conductive channel of $MoS_2$ lays across the PEDOT:PSS. The n-p-n junction device shows double Schottky rectifying characteristic (Supplementary Fig. 4d)[13].

The lower the surface potential is, the higher the work function is and the lower the electron concentration is. According to the relevant literature[14,15], the change of the $MoS_2$ electron concentration generally will bring about the fluctuation of the photoluminescence (PL) spectrum. Similar to its 2D surface potential image, PL intensity mapping of the device also depicts the triangle morphology of CVD monolayer $MoS_2$ (Fig. 1d). Besides, compared to the as-grown region, the PL spectrum intensity of the self-healed region was significantly enhanced, thus forming a clear dividing line between the as-grown and self-healed regions.

In fact, not only the PL spectrum intensity of the self-healed one is drastically enhanced, but also the peak energy is obviously blue shifted about 22 meV by PSS-induced SVSH (Fig. 1e). Previous studies reveal PL spectrum of monolayer $MoS_2$ is composed of A exciton and B exciton (~2.0 eV, purple). The prominent A exciton peak could be further evolved into exciton ($X_0$; ~1.86 eV; green) and trion ($X^-$; ~1.82 eV; blue) peaks. A negative trion is a quasiparticle composed of two electrons and a hole and formed through binding a neutral exciton (a photogenerated electron–hole pair) to an electron, the process consumes energy of ~40 meV (refs 8,16,17). By analysing the exciton peaks of the trion ($X^-$) and exciton ($X_0$), it explicitly manifests that the trion intensity is independent of PSS-induced SVSH (Fig. 1f). This phenomenon is ascribed to the large trion binding energy in monolayer $MoS_2$ (ref. 14). Simultaneously, the exciton intensity is almost twofold after PSS treatment, which is correlated with the decrease of the intrinsic heavy electron (n-type) doping in CVD $MoS_2$ (refs 14,17,18). This fact strongly suggests that the neutral excitons recombine rather than forming negative trions due to the decrease of the electron concentration. In our case, such electron concentration decrease is caused by the SVSH of the self-healed $MoS_2$. Our experimental results are consistent with the pronounced PL spectrum change induced by HBr treatment or gate doping[14,18]. In a word, from the PL enhancement, we can conclude that PSS-induced SVSH could dramatically tune the intrinsic electrons concentration, which is important for us to construct $MoS_2$ homojunction.

To further verify the PSS-induced self-healing effect and eliminate the PEDOT interference, the surface PSS of PEDOT:PSS film in device B was removed by 98% $H_2SO_4$ treatment[19,20], while the other device structure is unchanged. Neither the surface potential nor the intensity of the PL spectrum intensity changes between the overlapped and as-grown region in $MoS_2$ triangle (Supplementary Fig. 5b,c). Besides, Ohmic characteristic is observed between the overlapped $MoS_2$ and the PEDOT electrode (Supplementary Fig. 5d), indicating there is no work function difference between the overlapped and as-grown region. These experimental result suggests PEDOT itself has no impact on $MoS_2$ after PSS removal, and PEDOT:PSS-induced

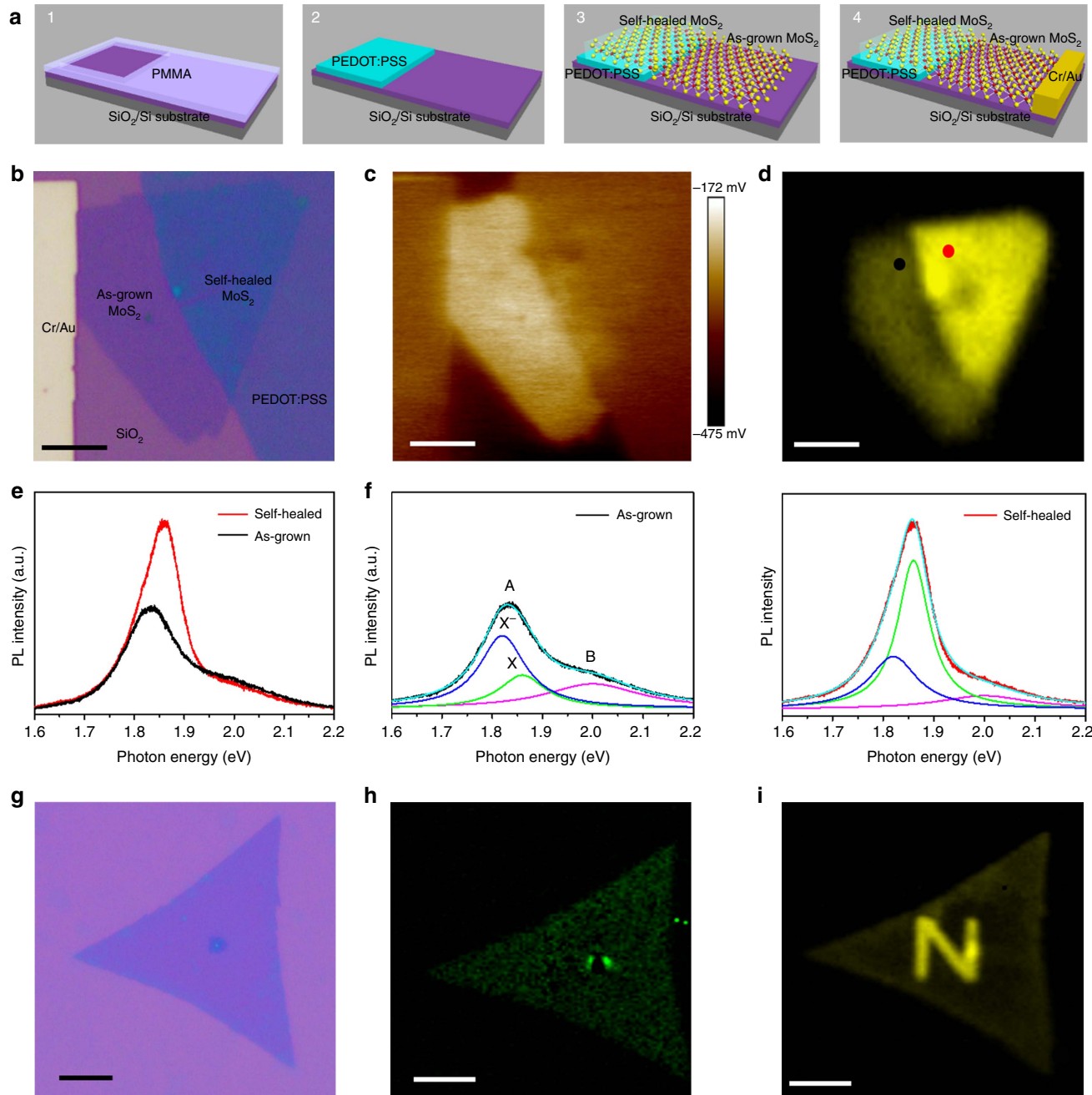

**Figure 1 | Complex characterization of sulfur vacancy self-healing.** (**a**) Construction process of the monolayer $MoS_2$ homojunction. (**b**) Optical microscopy (OM) image of the device $A_1$. Scale bar, $5\,\mu m$. (**c**) Corresponding 2D surface potential image. Scale bar, $5\,\mu m$. (**d**) Photoluminescence (PL) intensity mapping. Scale bar, $5\,\mu m$. (**e**) PL spectra acquired from different regions highlighted in **d**. (**f**) Comparison of the deconvoluted PL spectrum features in **e**. The experimental results are reproduced by the sum (cyan) of three peaks (trion $X^-$, blue; exciton $X_0$, green; exciton B, purple) assumed by Lorentzian functions. (**g–i**) OM image, Raman mapping constructed by integrating $E^1_{2g}$ mode and PL intensity mapping for a monolayer $MoS_2$ patterned by e-beam lithography (EBL) into the shape of the uppercase letter 'N' and PEDOT:PSS solution induced SVSH. Scale bar, $10\,\mu m$.

sulfur vacancy self-healing effect on $MoS_2$ does not originate from PEDOT but is derived from PSS.

On the basis of the above experimental results, PEDOT:PSS solution was also used to heal sulfur vacancies (Methods). PEDOT:PSS solution is a water-soluble solution and therefore is easily washed with water. The Raman spectra of the as-grown and self-healed $MoS_2$ before and after SVSH (Supplementary Fig. 6) did not change in the relative intensity or peak position. Thus, the structure of $MoS_2$ was not altered during healing, $MoS_2$ did not form any chemical bond with any other materials[21]. Meanwhile,

compared to the Raman spectrum (Supplementary Fig. 2a), PEDOT: PSS Raman peak was not found from the data, and we can obtain that there was no PEDSOT: PSS residue on the $MoS_2$ film surface. Besides, the frequency of $E^1_{2g}$ vibrational mode is sensitive to strain[22], Raman mapping of the $E^1_{2g}$ peak is quite uniform, indicating the lattice was not subjected to any induced strain from PEDOT:PSS (Fig. 1h). However, the letter 'N' was vividly engraved on a monolayer $MoS_2$ (Fig. 1i), and it attests to the advantages of our methodology for complicated pattern generation for monolithic system construction.

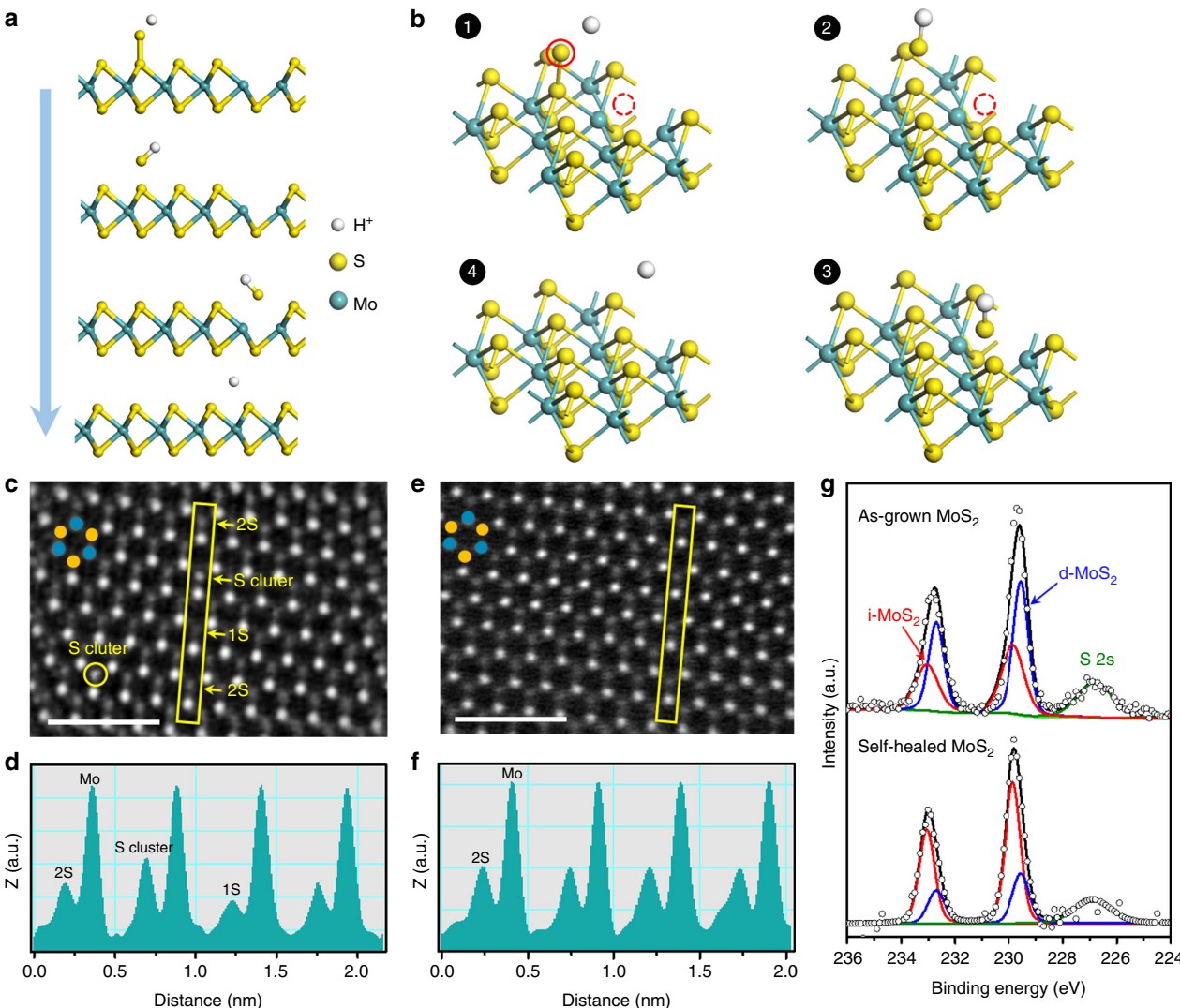

**Figure 2 | Sulfur vacancy self-healing (SVSH) mechanism.** (**a,b**) 2D/3D chemical structure change showing the PSS-induced SVSH effect. (**c–f**) The HAADF images before **c** and after **e** PSS-induced SVSH, together with the Z-contrast mapping done before **d** and after **f** in the areas marked with yellow rectangles, reveal that the sulfur vacancies (1S) are healed spontaneously by the sulfur adatom clusters on $MoS_2$ surface through a PSS-induced hydrogenation process. The cyan and yellow dots indicate the Mo and S atoms, respectively. Scale bar, 1 nm. (**g**) High-resolution XPS for Mo 3d before (top) and after (bottom) PSS treatment of $MoS_2$. Red and blue lines represent the intrinsic $MoS_2$ (i-$MoS_2$) and defective $MoS_2$ (d-$MoS_2$), respectively.

**Sulfur vacancy self-healing mechanism.** Thus the PSS-induced SVSH mechanism we proposed is that the hydrogenation of PSS guides sulfur adatom clusters on the as-grown $MoS_2$ surface to heal sulfur vacancies (Fig. 2a,b). Note that the sulfur vacancies are sufficiently shallow to act as electron donation defect in *n*-type monolayer $MoS_2$ (ref. 23). Contrary to sulfur vacancy, the sulfur adatom cluster is found to be an electrically neutral defect, even though its concentration is expected to be high[24]. In a word, the electrically neutral sulfur adatom clusters are used to fulfil the sulfur vacancies to precisely tune the electron concentration in monolayer $MoS_2$. Similar vacancy healing by superacid TFSI and hydracids (HCl, HBr, HI) has been used to enhance PL intensity of monolayer transition metal dichalcogenides (TMDs)[15,18,25]. However, as-mentioned overpowered acids corrode the metal electrodes, damage the electrode contact and display unsuitable in constructing homojunction. PSS shows great advantages in low-cost and mild acidic nature[26]. Moreover, dissociated polymers $PSS^-$ has so large molecular weight that it would not dope the sulfur vacancies and hinder the hydrogenation process[27].

To further confirm the SVSH mechanism, spherical aberration-corrected STEM was employed to obtain a direct vision of the atomic structure of the as-grown and self-healed $MoS_2$. Recently, the scanning transmission electron microscopy (STEM) technique has been proved to be powerful in providing comprehensive information of monolayer $MoS_2$ defects at the atomic scale. We visualized the films via chemical analysis using atomic-resolution Z-contrast imaging with high-angle annular-dark-field (HAADF) STEM. As the intensity of STEM images is directly related to the atomic number (Z-contrast)[9,18], sulfur vacancies (1S) and sulfur adatom clusters can be easily recognized and differentiated from the three-fold coordinated two sulfur atoms (Fig. 2c). The corresponding line profiles were extracted to give a clearer picture for sulfur atomic amounts (Fig. 2d). The three kinds of imaging contrasts, which corresponded to sulfur vacancies (1S), three-fold coordinated two sulfur atoms (2S) and sulfur adatom clusters, were presented obviously in the as-grown $MoS_2$. However, the self-healed $MoS_2$ displayed uniform intensity, which implied the PSS-induced SVSH could effectively decrease the sulfur vacancies and sulfur

adatom clusters (Fig. 2e). Other low magnification of the STEM images was displayed in Supplementary Fig. 7. We can draw the conclusion that the sulfur vacancies are healed spontaneously by the sulfur adatom clusters on $MoS_2$ surface through a PSS-induced hydrogenation process.

XPS was also used to identify whether sulfur vacancies were healed by the PSS-induced SVSH. The XPS spectra of Mo 3d consisted of two sets of peaks that can be respectively assigned to intrinsic $MoS_2$ (i-$MoS_2$) and defective $MoS_2$ (d-$MoS_2$) (Fig. 2g). The deconvoluted $Mo^{4+}$ $3d_{5/2}$ and $Mo^{4+}$ $3d_{3/2}$ doublet peaks depict the contributions of i-$MoS_2$ (doublets located at 232.70 and 229.55 eV) and d-$MoS_2$ (peaks at 233.05 and 229.85 eV). When the as-grown (as-grown) $MoS_2$ are healed by PEDOT:PSS solution treatment, the contribution of the intrinsic $MoS_2$ increases, whereas the defective $MoS_2$ component decreases. As a result, the doublets were shifted to the higher binding energy side. This reveals that the density of sulfur vacancies was diminished by the PSS treatment, as the d-$MoS_2$ peak is directly associated with sulfur vacancies[28]. In addition, The XPS spectra of S 2p also confirmed the sulfur vacancies healing (Supplementary Fig. 8). A similar behaviour was previously observed in sulfur vacancy healed $MoS_2$ through sulfurization annealing and molecular chemisorption[15,23,29,30]. To quantify the XPS information, we measured the XPS peak area ratio of S 2p to Mo 3d states for the as-grown and self-healed $MoS_2$. The value of S:Mo ratio was increased from $\sim 1.67$ to $\sim 1.86$ by the PSS-induced SVSH (Supplementary Note 1).

**Construction and electrical properties of $MoS_2$ homojunction.** I–V curve test was first conducted in another homojunction device $A_2$, which is on the basis of the device $A_1$ added with the EBL and the PEDOT: PSS solution induced SVSH process. The as-fabricated homojunction shows typical rectifying behaviour (Fig. 3b). To confirm the diode barrier was formed only in self-healed/as-grown junction, the electrical transport properties of other contact types were characterized. Ohmic characteristics are all observed among the following two contact types: the as-grown $MoS_2$ and Cr/Au electrode, and the self-healed $MoS_2$ and PEDOT: PSS electrode (Supplementary Fig. 9). So, the existence of potential barrier in other contact positions is excluded[31].

The rectifying performance of the homojunction is further quantitatively analysed under bias voltage by fitting to the diode equation[4,32–34]:

$$I_D = AA^* T^2 \exp\left(\frac{-q\varphi_B}{k_B T}\right)\left[\exp\left(\frac{qV_D}{nk_B T}\right) - 1\right], \quad (2)$$

where $A$ is the area of the Schottky junction, $A^*$ is the effective Richardson constant, $q$ is the elementary charge, $k_B$ is the Boltzmann constant, $T$ is the temperature, and $n$ is the ideality factor. Thus, the $n$ can be calculated from linearly fitting the natural logarithm plot of current and voltage, as depicted by the blue curve in Fig. 3a. Through equation (2), the ideality factor of our device is obtained as 1.6 from the black fitting line, which slightly deviate from the ideal value of 1. The reason is probably the large resistance of organic electrode PEDOT:PSS, which provides series resistance effect[7]. Quantitative analysis of the Schottky barrier height $\varphi_B$ can be done by investigating the temperature dependence of the diode current in the reverse bias saturation regime $(\exp(qV_D/nk_B T) \ll 1)$[35]. Here, the diode current becomes insensitive to $V_D$ and $I_{sat} \propto T^2 \exp(-q\varphi_B/k_B T)$. Figure 3c inset shows a plot of $\ln(I_{sat}/T^2)$ versus $q/k_B T$ in the reverse bias saturation regime. The Schottky barrier height $\varphi_B$ was estimated about 150 meV from the slope of the red curve.

The work function variation of the monolayer $MoS_2$ was also carefully double-checked by ultraviolet photoelectron spectroscopy (UPS). The work function can be calculated using[36,37]

$$\phi = hv - E_{onset}, \quad (3)$$

where $hv$ is the incident photon energy (20.22 eV) and $E_{onset}$ is the onset level related to the secondary electrons (Fig. 3d). Hence, the $\phi$ for the as-grown and self-healed $MoS_2$ is 4.35 and 4.55 eV, respectively. Note that the work function value obtained for the as-grown $MoS_2$ is consistent with several other reports[36,37]. The valence band ($E_v$) for the as-grown and self-healed $MoS_2$ is, respectively, located at 1.81 and 1.67 eV below the Fermi energy $E_F$ by linearly extrapolating the leading edge of the spectrum to the baseline (Fig. 3e). The work function difference between the as-grown and self-healed region is 140–200 meV, which was close to $\sim 150$ meV obtained by the variable temperature measurements of the homojunction diode behaviour. In addition, the optical band gaps of the CVD monolayer $MoS_2$ are determined to be $\sim 1.84$ eV from the PL spectrum (Fig. 1e). On the basis of the above results, the well-aligned energy band diagram, which has the same band gap but different Fermi level, is constructed to show the band bending behaviour at the interface of the as-grown and self-healed monolayer $MoS_2$ (Fig. 3f). The energy separation $\Delta E$ between the conduction band and $E_F$ of the self-healed $MoS_2$ is $\sim 170$ meV, indicating that self-healed monolayer $MoS_2$ is still n-doped. However, the energy separation $\Delta E$ in as-grown $MoS_2$ is only $\sim 30$ meV. Then, the as-grown $MoS_2$ region acted as an $n^+$ type, and the self-healed region acted as an n-type. An $n^+$-n monolayer $MoS_2$ homojunction was formed at the as-grown/self-healed junction.

We also investigated the effect of the SVSH on the electrical properties of a back-gated $MoS_2$ transistor at room $T$. Current decrease can be observed in the output characteristic curve and the threshold voltage dramatically shifted toward zero after the SVSH (Fig. 3g,h). The only constant is the Ohmic contact of the Au-$MoS_2$, which can be attributed to the transistor channel is long enough to ignore the changes in electrode contact. Besides, Supplementary Fig. 10 suggests the decrease of sulfur vacancies bring about the about 643 times decrease of electron concentration ranging from $5.56 \times 10^{19}$ to $8.65 \times 10^{16}$ cm$^{-3}$ (Supplementary Note 2), which can be comparable to the long-term sulfurization annealing[23]. These changes indicated that the electrons or sulfur vacancies in the as-grown $MoS_2$ was removed. An improvement in the subthreshold slope indicated that the SVSH reduces interface trap states. Similar phenomenon was previously observed in sulfur vacancy healed $MoS_2$ through sulfurization annealing and molecular chemisorption, and could be explained by a hopping transport model[23,29]. From another perspective, unipolar n-type electrical transport behaviour is observed in the self-healed $MoS_2$, which is consistent with the UPS measurements (Fig. 3d,e). Besides, the homojunction diode also behaves n-type behaviour (Supplementary Fig. 11), which again confirms the $n^+$-n homojunction structure.

In addition, the durability of the device also was investigated. As the PSS-induced SVSH is environmental-independent, the homojunction should be reliable under long-term operations. The rectifying behaviour or $I_{on}/I_{off}$ ratio of the homojunction have no degradation after two months storing under ambient conditions (Supplementary Fig. 12).

**The photovoltaic effect of $MoS_2$ homojunction.** The responsivity test of the photodiode was performed under variable incident light intensity (Fig. 4a). The as-fabricated homojunction shows an open circuit voltage of about 150 mV, which does not change significantly with different illumination power. The $\sim 150$ meV open circuit voltage is very close to the as-mentioned barrier height of variable temperature diode behaviour and UPS

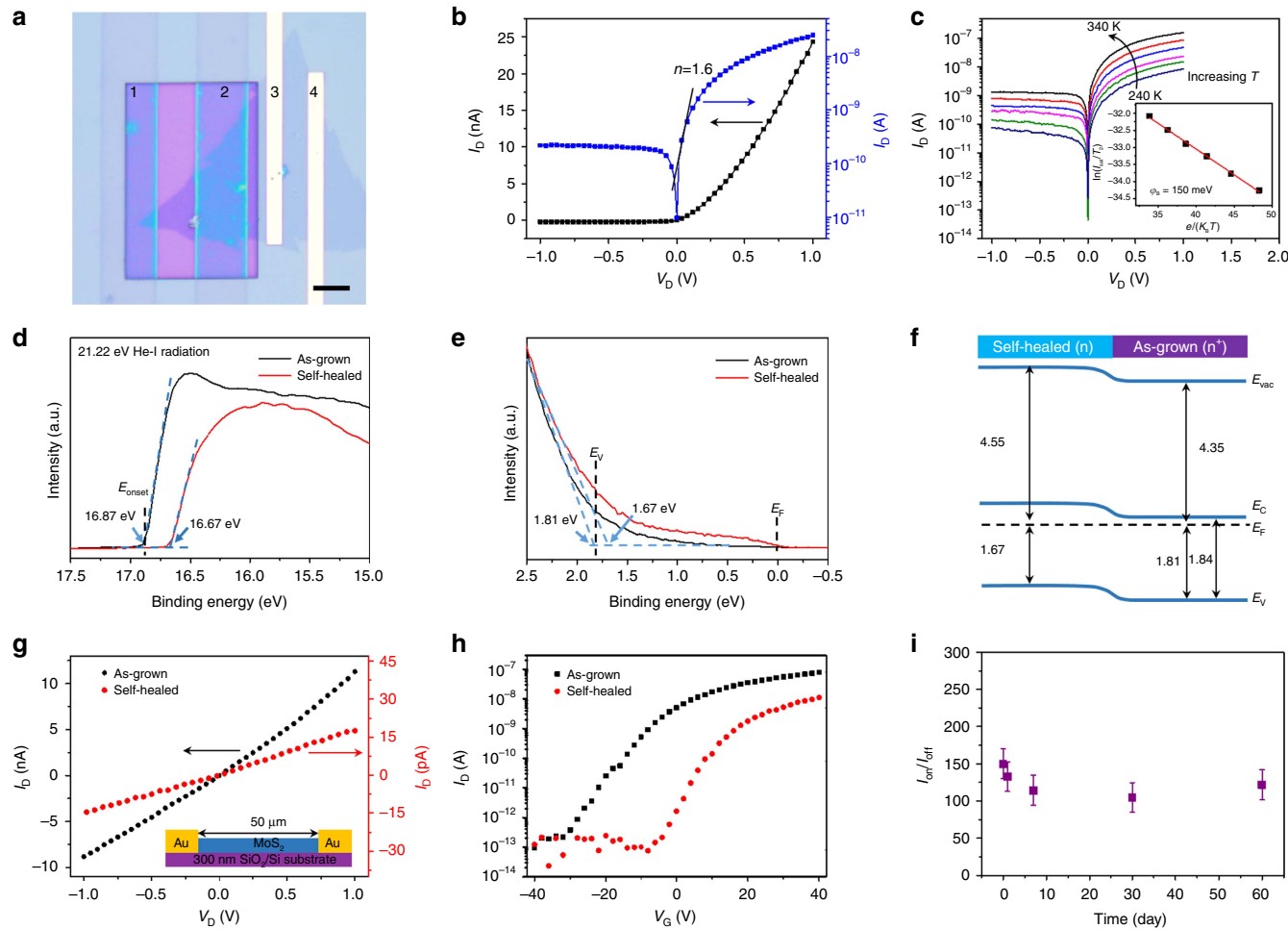

**Figure 3 | Construction and electrical properties of MoS$_2$ homojunction.** (**a**) OM image of the device A$_2$. PEDOT:PSS electrodes 1–2 define an self-healed MoS$_2$ FET, 3–4 define an as-grown MoS$_2$ FET, and 2–3 define the homojunction device A$_2$. Scale bar, 5 µm. (**b**) Output characteristic on linear/logarithmic scale (black/blue) of the device A$_2$. (**c**) Output characteristic of the homojunction under a series of temperatures. Inset: linear fitting result of the relationship between ln($I_{DS}/T^{3/2}$) and $e/(k_BT)$. The red line fit is drawn to yield the Schottky barrier height. (**d,e**) Secondary-edge and valence-band spectrum of the ultraviolet photoelectron spectroscopy (UPS) measurement from as-grown and self-healed monolayer MoS$_2$. (**f**) Band diagram of the monolayer MoS$_2$ homojunction obtained from UPS measurements. (**g,h**) Output characteristics and transfer characteristics of a monolayer MoS$_2$ transistor both before and after PSS-induced SVSH. (**i**) $I_{on}/I_{off}$ ratio of the homojunction measured during 60 days of storage under ambient conditions.

measurements. Certainly, the actual barrier height of our homojunction should be greater than this open-circuit voltage. Different from the open circuit voltage, the short circuit current increased with incident power (Fig. 4b). This indicates that the intensity of the light determines the number of photogenerated charge carriers, but not the homojunction band offset[9]. The responsivity decreases nonlinearly with the increasing light intensity, which is correlated to the decrease of unoccupied states in the conduction band of MoS$_2$ as light intensity increases[38]. The excellent responsivity of ~308 mA W$^{-1}$ at zero bias is much larger than that of other 2D homojunctions by chemical doping (Supplementary Table 1)[4,5,9,10,39,40]. It can be attributed to the wide space charge regions of the homojunction diode of ~150 meV barrier height.

## Discussion

The time-resolved photoresponse characteristics revealed a reliable photoresponse with a stabilized photocurrent ON/OFF ratio of ~200. In addition, the rise time (0–90%) and recovery time (10–100%) are 810 and 750 ms, respectively (Fig. 4c). The response speed is much faster than the CVD-grown MoS$_2$-based

photoconductive photodetectors[3,41,42], which generally have a long response time. To further explore the photoresponse origin, photocurrent map was performed to spatially investigate the local photoresponse. During this process, a focused laser beam is employed to illuminate a series of special points in the device, while the current is recorded as a function of position (Methods). The photocurrent maximum locates at the boundary between the self-healed and as-grown MoS$_2$, indicating the photoresponse arises from the homojunction rather than the MoS$_2$/PEDOT:PSS or MoS$_2$/metal contacts (Fig. 4d). In other words, the photogenerated electron–hole pairs are efficiently separated in the homojunction region and then transferred to the source and drain electrodes to generate photocurrent.

In conclusion, a lateral CVD monolayer MoS$_2$ homojunction was successfully fabricated by PSS-induced SVSH in selected region. We systematically proposed and demonstrated the self-healing mechanism, in which the sulfur vacancies (electron donation defect) are healed spontaneously by the sulfur adatom clusters (electrically neutral defect) on MoS$_2$ surface through a PSS-induced hydrogenation process. The SVSH preserved the original structure without additional local fields resulting the stable and efficient work function enhancement of ~150 meV. The electron concentration of

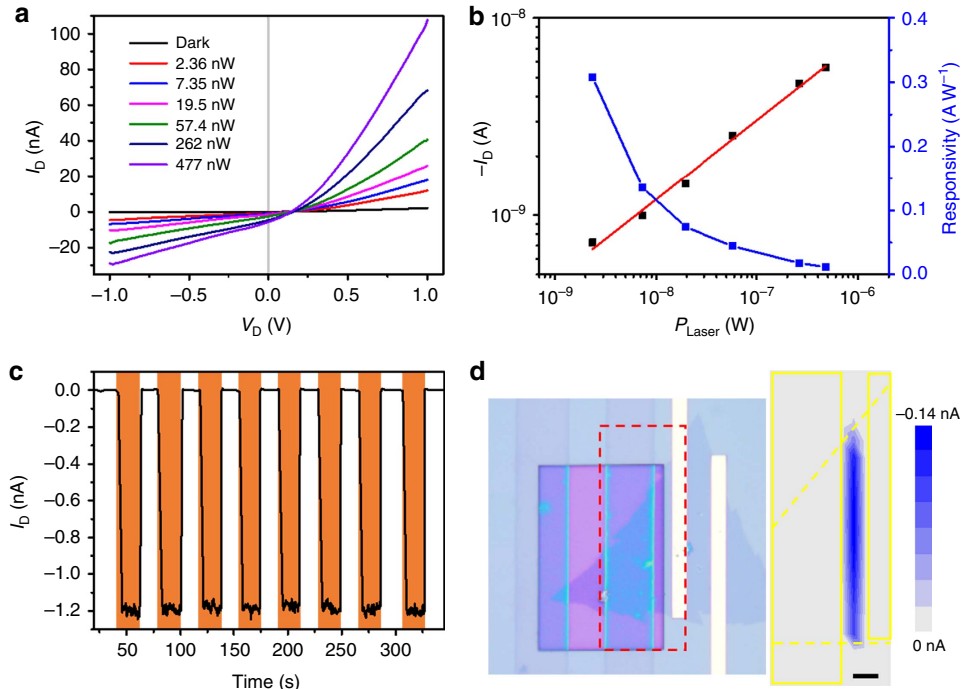

**Figure 4 | The photovoltaic effect of the MoS$_2$ homojunction.** (**a**) The photovoltaic effect of the monolayer MoS$_2$ homojunction under different 575 nm illumination intensities. (**b**) Dependence of photocurrent and responsivity on incident light intensity at zero bias. (**c**) Time-resolved photoresponse of the homojunction upon light illumination (575 nm, 19.5 nW) being turned on and off at zero bias. (**d**) Photocurrent map of the region indicated by the red rectangle in the OM image at zero bias. Scale bar, 2 μm. The largest photocurrent (blue region) originates from the boundary line of the as-grown and self-healed MoS$_2$ film.

the self-healed MoS$_2$ dramatically decreased by 643 times from $5.56 \times 10^{19}$ to $8.65 \times 10^{16}$ cm$^{-3}$. By using the SVSH process on an individual MoS$_2$, a homojunction was constructed at the interface of the as-grown and self-healed MoS$_2$. The diode presented perfect rectifying characteristic and excellent photoresponsivity of $\sim 308$ mA W$^{-1}$ at zero bias, which was much larger than that of other 2D homojunctions. The homojunction maintained an outstanding air-stability in the rectifying behaviour and photocurrent for more than two months. In addition, the cost-effective method showed more environment-independent than widely investigated chemical doping. Therefore, our findings paved a powerful strategy to control the 2D materials work functions and develop their homogeneous diodes for ultrathin, flexible, transparent and wearable electronic and optoelectronic nanodevices.

## Methods

**Growth of monolayer MoS$_2$.** This monolayer MoS$_2$ films were grown on the Si substrate with a 300 nm SiO$_2$ insulation layer by the chemical vapour deposition method. MoO$_3$ (Sigma-Aldrich, ≥99.5% purity) and sulfur (Sigma-Aldrich, ≥99.5% purity) were applied as precursor and reactant materials respectively. MoO$_3$ powder (25 mg) was placed in a quartz boat at the center of furnace. A $2 \times 2$ cm$^2$ SiO$_2$ substrates were put face down at top of the MoO$_3$ powder. S powder was heated to 180 °C by heating belt and carried through Ar flow of 500 s.c.c.m. The experiments were implemented at a reaction temperature of 850 °C for 30 min. Finally, the samples were taken out only if the furnace has naturally cooled down to room temperature.

**Monolayer MoS$_2$ transfer.** As-grown MoS$_2$ films were spin-coated with poly(methyl methacrylate) (PMMA) and submerged in 5% NaOH solution at 80 °C for 2 h. The PMMA/MoS$_2$ stacks were lifted from the solution, diluted in DI water, and then transferred onto target substrates only with PEDOT:PSS (Sigma-Aldrich, 1.0 wt%) or PEDOT. Subsequently, the substrates were annealed on the hotplate at 60 °C for 30 min to remove DI water and induce PSS to heal defects.

**PEDOT preparation by 98% H$_2$SO$_4$ treatment.** The substrate only with PEDOT:PSS electrode shown in Fig. 1a(2) was immersed into 98% H$_2$SO$_4$ for 15 min at room T, next sufficiently washed by DI water, and then dried at 100 °C for 10 min to remove residual DI water. Actually, there is still residual PSS$^-$ connected with PEDOT by hydrogen in 98% H$_2$SO$_4$ treated PEODT:PSS film[20], but for simplicity, the 98% H$_2$SO$_4$ treated PEDOT:PSS is referred to as PEDOT in this entire study.

**PEDOT:PSS solution induced sulfur vacancy self-healing.** Firstly, the MoS$_2$ sample was immersed in the PEDOT:PSS solution, after standing for 5 min, and then immersed in plenty of DI water to wash the PEDOT:PSS solution for 10 min. Further, the residual DI water was dried with nitrogen, finally the sample was dried at 100 °C for 10 min to remove the residual DI water of PEDOT:PSS electrode if the sample has PEDOT:PSS electrode.

**Measurements.** The KPFM measurements, AFM images and electrical curve of vertical junction were taken on a commercially available AFM (Nanoscope IIID, Multimode). The PL and Raman spectrum measurements were performed with a confocal microscopy (JY-HR800) under 514 nm laser with a power of 20 mW at room temperature. The spot size of the laser is about 1 μm$^2$. The step size for Raman and PL map is about 0.5 μm. All TEM samples were baked at 160 °C for 5 h under vacuum before the microscopy experiment. STEM imaging were performed on a JEM-ARM200F TEM. XPS was conducted with a Thermo Scientific ESCA Lab 250Xi XPS with a monochromatic KR Al X-ray line. Ultraviolet photoelectron spectroscopy (UPS) was performed in an ultrahigh vacuum chamber using a helium lamp source emitting (AXIS ULTRA DLD) at 21.2 eV. The photocurrent versus position curve used 514 nm laser as light source. The electrical characteristics and the photoresponse properties were implemented by a semiconductor analysis system (Keithley 4200). All electrical and optical signals were recorded in the ambient atmosphere, except variable temperature measurement.

**Data availability.** The data that support the findings of this study are available from the corresponding author on request.

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

## Acknowledgements

This work was supported by the National Major Research Program of China (No. 2013CB932602), the National Key Research and Development Program of China 2016YFA0202701, the Program of Introducing Talents of Discipline to Universities (B14003), the National Natural Science Foundation of China (Nos 51527802, 51232001, 51602020, 51672026 and 51372020), China Postdoctoral Science Foundation (2015M580981, 2016T90033) Beijing Municipal Science & Technology Commission, and the State Key Laboratory for Advanced Metals and Materials (No. 2016Z-06), the Fundamental Research Funds for the Central Universities (FRF-TP-15-075A1, FRF-BR-15-036A, FRF-AS-15-002).

## Author contributions

X.Z. and S.Z. deposited MoS₂ films by CVD. X.Z., S.L. and Z.Z. performed the device fabrication, data collection and analysis. J.D. assisted in carrying out the film fabrication and characterizations, and J.X. and Z.K. assisted in the device performance measurements. F.L. and B.L. carried out KPFM and AFM measurements. X.Z., Z.K. and Y.O. performed part of the Raman, PL and UPS characterization. K.G. and X.L. carried out STEM experiments. X.Z., Q.L., Z.Z. and Y.Z. initiated and supervised the project. All authors discussed the results, and prepared and commented on the manuscript.

## Additional information

**Competing interests:** The authors declare no competing financial interests.

