## [Peer Review File · Nature Communications]

Reviewers' comments:

Reviewer #1 (Remarks to the Author):

The manuscript describes an approach to a chemical modification of MoS₂ grown by powder vaporization. Modification has the following effects:

1. Change in the ratio of sulfur XPS peaks associated with defective versus intact sites.
2. Increase in overall photoluminescence intensity by a factor of two, and an increase in the intensity of the neutral A exciton by a factor of approximately 4.
3. An increase in work function of ~58 meV.
4. Weak rectification is observed in a homojunction created by chemical modification.
5. The observed effects persist for two months.

A means to make MoS₂ p-type and stable with reasonable mobility would represent a significant advance. Despite the range of materials and device characterization presented in the manuscript, and the demonstration of an n-n+ homojunction, it does not represent a major advance.

1. Repair of defects should improve mobility. Mobility is not analyzed, and it does not appear to be high based on the currents achieved in the devices.
2. The diode analysis is not convincing. There is still a high density of defects and trap states (obvious from the photoresponse) that seem inconsistent with an ideality factor of 1. The diode behavior, which is analyzed over too small a range, is clearly not ideal. The analysis does not explicitly account for a significant series resistance, and the breakdown in reverse bias is not observed, though it should be readily observed with such a small barrier.
3. Furthermore, the band diagram in figure 4 is not correct. The band bending at the homojunction, which should also be the built-in voltage, is significantly greater than half of the bandgap. But 58 meV is much less than half the bandgap. The Fermi level is also shown close to the valence band, presumably to match the work function measurements, but the transport data (Fig. 6a) do not suggest a Fermi level shift of that magnitude.
4. I'm skeptical that the Fermi level actually shifts (from being pinned near the conduction band) to the extent that the work function matches that of PEDOT:PSS. The surface potential measurement is an excellent experiment to do in this context, but the analysis neglects the possibility that the substrate directly impacts the measurement due to incomplete screening in the MoS₂. This seems a likely source of the variation, regardless of the fact that the PSS does modify the MoS₂. (Also, the figure panels are mislabeled or switched in figure 3).
5. The extremely slow photoresponse indicates that this is not a useful photodetector. Ultimately, the large number of device measurements do not add significant fundamental understanding of how the material is modified by the chemical treatment.

The study is fundamentally interesting and aspects are potentially worth publishing if more fundamental insight into the actual mechanism and impact of the material could be provided. For example, the increased magnitude of the photoresponse at the homojunction is interesting. The bias dependence could be used to help develop correct band diagrams as a function of bias. A direct comparison and analysis of transistor performance with and without modification would be informative. I would be interested to see whether one can develop a self consistent picture of changes in carrier concentration and mobility, and whether the parasitic capacitance due to traps might be analyzed by variable temperature measurements. Variable temperature measurements of the homojunction diode behavior are also recommended to provide a more convincing and complete analysis.

Reviewer #2 (Remarks to the Author):

I have reviewed "A monolayer MoS₂ homojunction construction strategy: PSS-induced sulfur vacancy self-healing" written by Zhang et al. The work is interesting, and the material and

electrical characterizations are well performed. However, I think that the data presented and interpretation made to draw conclusion are premature to publish in Nature Communications. See the following comments.

- In this work, MoS₂ was prepared by CVD. There can be many external defects originated from CVD process, eg. grain boundary. I think that these external defects related to poly-crystalline structure need to be distinguished from internal sulphur vacancies as intrinsic lattice defects of single crystalline MoS₂. If the authors show the comparison results with exfoliated MoS₂, this issue may be explained better.
- The authors describe "self-healing process" briefly as "The self-healing mechanism is that the sulfur vacancies are healed spontaneously by the sulfur adatom clusters on MoS₂ surface through a PSS-induced hydrogenation process." I suggest them to provide a more systematic model in the atomic level, on how the clusters are disintegrated and sulfur adatoms can make surface migration to form bonding at vacancies.
- It may be desired to show energy consideration on self-healing, eg. formation energy of sulfur adatoms to S-Mo bonding. Can PSS-induced hydrogenation provide such energy?
- The Fermi level shift of 58 meV is considered small, to induce polarity change of MoS₂. Do the KPFM results provide reliable correlation to electron concentration? Although the proposed formulas (1) – (3) look reasonable, I am puzzled on how such small change can induce a good homojunction.
- With the 58meV built-in potential in the homojunction which likely is widened by diffusional effect at the junction, I am reluctant to accept the formation of efficient pn junction. If the authors can provide evidences on the shift of Fermi level other than KPFM, the data can be explained more reasonably. What about trying additional ARPES measurement?
- The authors claim air-stability of 2 months for the proposed doping. Only one data set related to this claim is shown in Fig S6. It will be better if more supporting data are provided. For example, more temporal data after 1 day, 1 week, etc. for various light illuminations.

Reviewer #3 (Remarks to the Author):

The authors demonstrated a surface treatment method to tune the electronic property of monolayer MoS₂ by providing PSS acid to the surface for healing the sulfur vacancies. A homojunction can be formed between untreated and PSS-treated monolayer MoS₂, exhibiting stable photodiode performance. This work is interesting. However, if the authors claim the sulfur vacancy healing resulting from PSS treatment, they should provide more solid evidences to confirm the conclusion and quantify the change value of the sulfur vacancies induced from the PSS-treatment. The authors should also clarify more the difference in their surface treatment for healing sulfur vacancies, compared to other reported surface treatment methods. Therefore, I would not recommend a publication to Nature Communications. Several concerns are followed:

1. The authors should provide more solid evidence to confirm their conclusion: "Surface acid treatment healing can penetrate easily through the entire flake."
2. The authors ascribed an ideal n value observed in PSS-treated MoS₂ junction to less interface charge traps, compared to conventional p-n heterojunctions. They should clarify why there are less charge traps at this interface, since PSS-treatment caused a decrease of electron concentration by decreasing the sulfur vacancies.
3. In Figure 3C, the difference of work function between the overlapped MoS₂ and Cr/Au is more than 100 meV. And in Figure 3F, the difference of work function between untreated MoS₂ and Cr/Au is 58 meV. Why there is a big difference in those two contact areas.
4. Is there a possible method to tune this work function difference between PSS-treated and

untreated MoS₂ during the PSS-treatment process?

5. What's the electron concentration in the PSS-treated MoS₂?

6. The authors should clarify more Why there is a bigger gap between the $V_g=10V$ and $0V$, compared to other gating voltage curves in the $I_{ds}-V_{ds}$ plots in Figure 4b.

7. Why the source-drain current becomes hard to be tuned by negative gating as well as positive voltages as shown in the Figure 6a?

8. What is the response behavior of the untreated MoS₂ junction with /without light illumination? After a discharge, what would happen to the current decay time in an untreated MoS₂ junction.

Reviewers' comments (NCOMMS-16-23106):

Reviewer #1:

The manuscript describes an approach to a chemical modification of MoS₂ grown by powder vaporization. Modification has the following effects:

1. Change in the ratio of sulfur XPS peaks associated with defective versus intact sites.
2. Increase in overall photoluminescence intensity by a factor of two, and an increase in the intensity of the neutral A exciton by a factor of approximately 4.
3. An increase in work function of ~58 meV.
4. Weak rectification is observed in a homojunction created by chemical modification.
5. The observed effects persist for two months.

A means to make MoS₂ p-type and stable with reasonable mobility would represent a significant advance. Despite the range of materials and device characterization presented in the manuscript, and the demonstration of an n-n⁺ homojunction, it does not represent a major advance.

1. Repair of defects should improve mobility. Mobility is not analyzed, and it does not appear to be high based on the currents achieved in the devices.

A: We thank the reviewer for raising this question. It's a good point. Now we investigate the effect of the sulfur vacancy self-healing (SVSH) on the electrical properties of a back-gated MoS₂ transistor at room *T*. Current decrease can be observed in the output characteristic curve (**new Figure 3g**), and the threshold voltage (**new Figure 3h**), dramatically shifted toward zero after the sulfur vacancy self-healing. These changes indicated that the electrons or sulfur vacancies in the as-grown MoS₂ was removed. An improvement in the subthreshold slope indicated that the SVSH reduced interface trap states. In fact, the transistor conductive channel is as long as 50 μm so that the Ohmic contact of the Au-MoS₂ has not been changed shown in **new Figure 3g**.

New figure 3g-h, Output and transfer characteristics of a monolayer MoS₂ transistor, both before and after PSS-induced sulfur vacancy self-healing.

Figure R1. Transfer characteristics of a monolayer MoS₂ transistor, both before and after treatment. V_{DS}, drain-source voltage; S, source; D, drain; G, gate. Reproduced from reference [Science 350, 1065-1068 (2015)].

In the classical semiconductor physics, materials have less defects, the smaller the electron concentration, the lower the scattering probability and the higher the mobility. However in MoS₂ films, lowering the concentration of sulfur vacancies indeed decreases the electron concentration, but its performance is still declining mobility. This can be explained by the hopping transport model [Nat. Commun. 4, 2642 (2013)], in which electrons in the MoS₂ transport through the sulfur vacancies by hopping process. Through this model, the average distance between the sulfur vacancies would increase by PSS-induced SVSH. Therefore, it will make both the hopping probability and mobility decrease. Similar behavior was previously observed in sulfur vacancy healed MoS₂ through sulfurization annealing and molecular chemisorption [ACS nano 8, 10551-10558 (2014) and ACS nano 9, 8044-8053 (2015)].

2. The diode analysis is not convincing. There is still a high density of defects and trap states (obvious from the photoresponse) that seem inconsistent with an ideality factor of 1. The diode behavior, which is analyzed over too small a range, is clearly not ideal. The analysis does not explicitly account for a significant series resistance, and the breakdown in reverse bias is not observed, though it should be readily observed with such a small barrier.

A: We appreciate the reviewer for discovering this mistake. the I–V curve of diode consists of four regions: I) reverse bias region, II) linear diode region, III) current injection region, and IV) series resistance dominant region [Adv. Mater. 2015, 27, 5534–5540]. The ideality factor (n) should be estimated at forward bias (linear diode region). It was re-calculated in the linear diode region and now shown in **new Figure 3b**. Following this comments, the voltage sweep area is extended to -20~20 V shown in **new Figure S11a**. The breakdown voltage in reverse bias of the homojunction is about -12 V, indicating the small barrier of the homojunction does show a very stable characteristic.

New Figure 3b. I-V characteristic of the device A_2 on linear/ logarithmic (black/blue) scale, showing a diode behavior. The ideality factor is obtained as 1.6 from the black fitting line.

New Figure S11a. I-V curve of the monolayer MoS_2 homojunction. The breakdown voltage in reverse bias is about -12 V, which can be attributed to the small barrier of the MoS_2 homojunction.

3. Furthermore, the band diagram in figure 4 is not correct. The band bending at the homojunction, which should also be the built-in voltage, is significantly greater than half of the bandgap. But 58 meV is much less than half the bandgap. The Fermi level is also shown close to the valence band, presumably to match the work function measurements, but the transport data (Fig. 6a) do not suggest a Fermi level shift of that magnitude.

A: We appreciate the reviewer for discovering this mistake. A new band structure (new Figure 3f) is used to replace the old one. UPS (ultraviolet photoelectron spectroscopy)

measurement was used to measure the work function and the valence band of monolayer MoS₂ before and after the PSS-induced SVSH. The work function difference between the as-grown and self-healed region is 140~200 meV, which was close to the ~150 meV barrier height obtained by variable temperature electrical measurements of the homojunction diode.

New Figure 3d-f. d-e, Secondary-edge and valence-band spectrum of the UPS measurement from as-grown and self-healed monolayer MoS₂. f, Band diagram of the monolayer MoS₂ homojunction obtained from UPS measurements.

4. I'm skeptical that the Fermi level actually shifts (from being pinned near the conduction band) to the extent that the work function matches that of PEDOT:PSS. The surface potential measurement is an excellent experiment to do in this context, but the analysis neglects the possibility that the substrate directly impacts the measurement due to incomplete screening in the MoS₂. This seems a likely source of the variation, regardless of the fact that the PSS does modify the MoS₂. (Also, the figure panels are mislabeled or switched in figure 3).

A: We thank the reviewer's comment. To analyze the Fermi level shifts, two key issue was considered in our revised manuscript, including the influence of substrate and a more convincing measurement to support the result. Firstly, we consider the screening influence of substrate on surface potential measurements. When the surface PSS of PEDOT:PSS electrode is removed, there is no surface potential (work function) variation between the overlapped and as-grown MoS₂ triangle shown in **Figure R2b**. However, in the **Figure R2d**, the apparent brightness difference appears near the boundary between the self-healed and the as-grown MoS₂ triangle shown. The experimental result suggests PEDOT itself has no impact on MoS₂ after PSS removal, PSS is a major factor affecting the work function of monolayer MoS₂.

Figure R2. From top (without PSS treatment) to bottom (with PSS treatment), (a,c) optical microscopy images, (b,d) corresponding two-dimensional (2D) surface potential images.

New Figure 3d-e. Secondary-edge and valence-band spectrum of the UPS measurement from as-grown and self-healed monolayer MoS₂.

On the other hand, the PEDOT:PSS substrate itself for MoS₂ work function change is not necessary. As shown in **new Figure 3d-e**, the work function and valence band of monolayer MoS₂ were significantly changed by PEDOT:PSS solution healing. PEDOT:PSS solution can be easily removed by DI water. Certainly, different from the surface potential (58 meV) measured at the atmosphere, the work function variation of UPS measurement measured in ultrahigh vacuum environment is ~200 meV.

Besides, the frequency of E_{2g}^1 vibrational mode is sensitive to strain, Raman mapping of the E_{2g}^1 peak is quite uniform, indicating the lattice was not subjected to any induced strain (**Fig. 1h**). However, the letter “N” was vividly engraved on a monolayer MoS₂ (**Fig. 1i**), and it attests to the advantages of our methodology for complicated pattern generation for monolithic system construction.

New Figure 1g-i, OM image, Raman mapping constructed by integrating E_{12g} mode and PL intensity mapping for a monolayer MoS₂ patterned by e-beam lithography (EBL) into the shape of the uppercase letter “N” and PEDOT:PSS solution induced SVSH.

5. The extremely slow photoresponse indicates that this is not a useful photodetector. Ultimately, the large number of device measurements do not add significant fundamental understanding of how the material is modified by the chemical treatment.

A: We thank the reviewer’s comment. When the device structure was made some minor changes, the rise time (the time taken by the PD to reach 90% of the maximum photocurrent from the dark current) and recovery time (the time taken to reach 10 % of the maximum photocurrent) are reduced to 810 ms and 750 ms at zero bias, respectively (**new Figure 4c**).

The response speed is much faster than the MoS₂ based photoconductive photodetectors, which generally have a long response time.

For another question, we added STEM, UPS, PL mapping, mobility change and variable temperature homojunction rectifying behavior to further study the PSS-induced sulfur vacancy self-healing mechanism.

New figure 4c, Time-resolved photoresponse of the homojunction upon light illumination (575 nm, 19.5 nW) being turned on and off at zero bias

The study is fundamentally interesting and aspects are potentially worth publishing if more fundamental insight into the actual mechanism and impact of the material could be provided. For example, the increased magnitude of the photoresponse at the homojunction is interesting. The bias dependence could be used to help develop correct band diagrams as a function of bias. A direct comparison and analysis of transistor performance with and without modification would be informative. I would be interested to see whether one can develop a self consistent picture of changes in carrier concentration and mobility, and whether the parasitic capacitance due to traps might be analyzed by variable temperature measurements. Variable temperature measurements of the homojunction diode behavior are also recommended to provide a more convincing and complete analysis.

A: We thank for the reviewer's positive suggestions. For the change in electron concentration and work function, various methods were used to confirm the results. At the same time, the MoS₂ homojunction constructed by the PSS-induced SVSH displays photovoltaic effect shown in new **Figure 4a** and **4d**.

New Figure 4a, The photovoltaic effect of the monolayer MoS₂ homojunction under different 575 nm illumination intensities.

New Figure 4d, Photocurrent map of the region indicated by the red rectangle in the OM image at zero bias. The largest photocurrent (blue region) originates from the boundary line of the as-grown and self-healed MoS₂ film.

Reviewer #2 (Remarks to the Author):

I have reviewed "A monolayer MoS₂ homojunction construction strategy: PSS-induced sulfur vacancy self-healing" written by Zhang et al. The work is interesting, and the material and electrical characterizations are well performed. However, I think that the data presented and interpretation made to draw conclusion are premature to publish in Nature Communications. See the following comments.

1. In this work, MoS₂ was prepared by CVD. There can be many external defects originated from CVD process, eg. grain boundary. I think that these external defects related to polycrystalline structure need to be distinguished from internal sulphur vacancies as intrinsic lattice defects of single crystalline MoS₂. If the authors show the comparison results with exfoliated MoS₂, this issue may be explained better.

A: We thank the reviewer's comment. For the acid induced sulfur vacancies healing with its sulfur clusters, Matin Amani have used organic superacid TFSI to repair the sulfur vacancies of the exfoliated MoS₂ film [Science 350, 1065-1068 (2015)]. Besides, Hau-Vei Han used hydrohalic acid HBr to repair the CVD monolayer MoS₂ [ACS nano 10, 1454-1461 (2016)]. However, as-mentioned overpowered acids corrode the metal electrodes, damage the electrode contact and display unsuitable in constructing homojunction. More seriously, in the repairing process, Br ions will be used to fill the sulfur vacancies.

2. The authors describe "self-healing process" briefly as "The self-healing mechanism is that the sulfur vacancies are healed spontaneously by the sulfur adatom clusters on MoS₂ surface through a PSS-induced hydrogenation process." I suggest them to provide a more systematic model in the atomic level, on how the clusters are disintegrated and sulfur adatoms can make surface migration to form bonding at vacancies.

A: We thank the reviewer's suggestion. To investigate the mechanism of the self-healing, spherical aberration-corrected STEM has been employed to obtain a direct vision of the atomic structure of the as-grown and self-healed MoS₂. As shown in new Figure 2c-f, sulfur clusters and sulfur vacancies in CVD monolayer MoS₂ can be easily found, but after the PEDOT:PSS solution sulfur vacancies self-healing, we do not see these in as-healed MoS₂. Combined with the experimental data from XPS, we confirm that PSS-induced sulfur vacancy self-healing facilitates a shift in monolayer MoS₂ towards the intrinsic (defect-free) structure.

New Figure 2c-f. The HAADF images before (c) and after (e) PSS-induced SVSH, together with the Z-contrast mapping done before (d) and after (f) in the areas marked with yellow rectangles, reveal that the sulfur vacancies are healed spontaneously by the sulfur adatom clusters on MoS₂ surface through a PSS-induced hydrogenation process. The red and yellow dots indicate the Mo and S atoms, respectively.

3. It may be desired to show energy consideration on self-healing, eg. formation energy of sulfur adatoms to S-Mo bonding. Can PSS-induced hydrogenation provide such energy?

A: We thank the reviewer's suggestion. According to the data presented in **new Figure R3**, we can extract the formation energy of S interstitial (S_i, also called sulfur adatoms) of ~1 eV is less than the S vacancy (V_S) of ~3 eV. The researches in this regard are only rarely reported, on the other hand, our area of expertise is not in this chemicals but in the physical field, so this issue is not well described.

Figure R3. Calculated formation energies of the isolated native defects of S vacancy (V_S), S interstitial (S_i), Mo vacancy (V_{Mo}), and Mo interstitial (Mo_i) in a monolayer MoS_2 as a function of the Fermi level in the S-rich limit conditions [PHYSICAL REVIEW B 89, 205417 (2014)].

4. The Fermi level shift of 58 meV is considered small, to induce polarity change of MoS_2 . Do the KPFM results provide reliable correlation to electron concentration? Although the proposed formulas (1) – (3) look reasonable, I am puzzled on how such small change can induce a good homojunction.

With the 58meV built-in potential in the homojunction which likely is widened by diffusional effect at the junction, I am reluctant to accept the formation of efficient pn junction. If the authors can provide evidences on the shift of Fermi level other than KPFM, the data can be explained more reasonably. What about trying additional ARPES measurement?

A: We thank the reviewer's comment. Following this comment, we calculate the barrier height (the work function change) of the homojunction through the rectifying curve of under different temperatures, as shown in **new Figure 3c**. The Schottky barrier height ϕ_B was estimated as about 150 meV from the slope of the red curve. Besides, the work function variation of the monolayer MoS_2 was also double-checked by the UPS (ultraviolet photoelectron spectroscopy) results in **Figure R3**. The work function difference between the as-grown and self-healed region is 140~200 meV, which is close to the result obtained by electrical measurements in **new Figure 3c**.

New Figure 3c. Output characteristic of the homojunction under a series of temperatures. Inset: linear fitting result of the relationship between $\ln(I_{D}/T^{3/2})$ and $e/(k_{B}T)$. The Schottky barrier height can be extracted from the slope of the red line.

New Figure 4a. The photovoltaic effect of the monolayer MoS₂ homojunction under different 575 nm illumination intensities.

At the same time, the responsivity test of the photodiode was performed under variable incident light intensity, as shown in **new Figure 4a**. The as-fabricated homojunction shows an open circuit voltage of about 150 mV, which does not change significantly with different illumination power. The 150 meV open circuit voltage is very close to the as-mentioned barrier height of variable temperature diode result and UPS measurements. Indeed, as the reviewer said, monolayer MoS₂ don't display polarity change, which was proved by the n-type transfer curve of the self-healed MoS₂ transistor in **new Figure 3h**. Although we did not find a place to complete ARPES measurements, but we believe that the current performance of the variable-temperature diode rectifying behavior and UPS measurements are sufficient to calculate the barrier height (work function change).

New Figure 3h. Transfer characteristics on the logarithmic scale of a monolayer MoS₂ transistor, both before and after PSS-induced sulfur vacancy self-healing.

5. The authors claim air-stability of 2 months for the proposed doping. Only one data set related to this claim is shown in Fig S6. It will be better if more supporting data are provided. For example, more temporal data after 1 day, 1 week, etc. for various light illuminations.

A: We thank the reviewer's suggestion. Following this comments, we have made some modifications to the device structure to fulfill the research needs. As shown in new Figure S11b, the output curve of the homojunction was obtained in temporal data. The I_{on}/I_{off} ratio of the homojunction shows good stability. In spite of the damage of the light source, we are not able to obtain good photoresponse stability of the later homojunction.

New Figure S11b and 3i. Output curve and I_{on}/I_{off} ratio of the homojunction measured during 60 days of storage under ambient conditions.

Reviewer #3 (Remarks to the Author):

The authors demonstrated a surface treatment method to tune the electronic property of monolayer MoS₂ by providing PSS acid to the surface for healing the sulfur vacancies. A homojunction can be formed between untreated and PSS-treated monolayer MoS₂, exhibiting

stable photodiode performance. This work is interesting. However, if the authors claim the sulfur vacancy healing resulting from PSS treatment, they should provide more solid evidences to confirm the conclusion and quantify the change value of the sulfur vacancies induced from the PSS-treatment. The authors should also clarify more the difference in their surface treatment for healing sulfur vacancies, compared to other reported surface treatment methods. Therefore, I would not recommend a publication to Nature Communications. Several concerns are followed:

A: We thank the reviewer's comment. In this revised manuscript, we add systematical and fundamental experiments to support our conclusions. For example, the electron concentration of the self-healed MoS₂ dramatically decreased by 643 times, leading to the work function enhancement up to ~150 meV. Compared to other reported surface treatment methods, healing intrinsic lattice defects is a fundamental and efficient approach to control the work function without introducing additional local fields. Both chemical doping and thermal annealing will introduce the additional local fields or Coulomb's scattering sites.

1. The authors should provide more solid evidence to confirm their conclusion: "Surface acid treatment healing can penetrate easily through the entire flake."

A: We appreciate the reviewer for discovering this mistake. We are sorry that this sentence made the reviewer misunderstand. This sentence has been deleted. What we wanted to explain is the method of the self-healing is facile and stable.

2. The authors ascribed an ideal n value observed in PSS-treated MoS₂ junction to less interface charge traps, compared to conventional p-n heterojunctions. They should clarify why there are less charge traps at this interface, since PSS-treatment caused a decrease of electron concentration by decreasing the sulfur vacancies.

A: We thank the reviewer's comment. Some pervious literatures show that the increase of carrier traps density in the diodes leads to enhance the ideal factor of the diodes as a consequence of carriers recombination processes [Sol. Energy Mater. Sol. Cells 2000, 62 , 393 Physics of Semiconductor Devices , Wiley, Hoboken, NJ, USA 2007 and Adv. Mater. 2015, 27, 5534–5540]. In general, a heterojunction is made up of p-type and n-type materials, and in the building process, the lattice dislocations and impurities will be inevitably introduced, resulting in the generation of a local field (or charge trap) which reduces the electrical properties of the heterojunction. Distinguished with previous methods, the lattice defects induced local fields are eliminated during the process of the sulfur vacancy self-healing, which largely improve the performance of MoS₂ homojunction. As shown in **new Figure 3b**, the MoS₂ homojunction diode displays perfect rectifying behavior. Besides, the rectifying behavior of variable temperature measurement show that the Schottky barrier height is 150 meV (**new Figure 3b**). The result is close to the value obtained from UPS measurements.

New figure 3b and 3c, **b**, Output characteristic on linear/logarithmic scale (black/blue) of the device A₂. **c**, Output characteristic of the homojunction under a series of temperatures. Inset: linear fitting result of the relationship between $\ln(I_D/T^{3/2})$ and $e/(k_B T)$. The red line fit is drawn to yield the Schottky barrier height.

New Figure 3d-f. d-e, Secondary-edge and valence-band spectrum of the UPS measurement from as-grown and self-healed monolayer MoS₂. f, Band diagram of the monolayer MoS₂ homojunction obtained from UPS measurements.

3. In Figure 3C, the difference of work function between the overlapped MoS₂ and Cr/Au is more than 100 meV. And in Figure 3F, the difference of work function between untreated MoS₂ and Cr/Au is 58 meV. Why there is a big difference in those two contact areas.

A: We thank the reviewer's comment. The details we need to point out is that the device in

Figure 3c is not the same as the device in **Figure 3f**. We may have different levels of residue in the removal of PMMA of EBL. The measurement of the surface potential is affected by the presence of PMMA residue on the MoS₂ or Cr/Au surface. The contact potential difference between two materials depends on a variety of parameters such as the work function, adsorption layers, oxide layers, dopant concentration in semiconductors, or temperature changes of the sample [RSC Adv. 5, 42075-42080 (2015) and Nanoscale 7, 4461-4467 (2015)]. Surely, there may be difference between the electron concentrations of two MoS₂ films.

On the other hand, to exclude the disturbance of the PMMA residue and MoS₂ films sample, the work function MoS₂ film before and after PEDOT:PSS solution induced sulfur vacancy self-healing was double checked by UPS measurement. Notably, the two samples was a same MoS₂ film, the only difference is the PEDOT:PSS solution induced sulfur vacancy self-healing. The only difference is whether the MoS₂ film sample is healed by PEDOT:PSS solution induced SVSH.

4. Is there a possible method to tune this work function difference between PSS-treated and untreated MoS₂ during the PSS-treatment process?

A: We thank the reviewer's suggestion. As far as we concerned, we think we can tune the concentration of the PEDOT:PSS solution, healing time and other ways to change the degree of PSS-induced sulfur vacancy self-healing, so as to achieve the purpose of adjusting the work function difference. These will be studied by our subsequent work in the near future. Now, the focus of this research is obviously not the work function difference control.

5. What's the electron concentration in the PSS-treated MoS₂?

A: We thank the reviewer's comment. The effect of the SVSH on the electrical properties of a back-gated MoS₂ transistor at room T was investigated. The channel of the monolayer transistor effectively behaves as a resistor with conductivity $\sigma = q\mu N_D$, and the conductivity can also be calculated using the expression $\sigma = \frac{1}{\rho} = 1/(\frac{dV_D}{dI_D} \times \frac{WH}{L})$, where N_D is the electron concentration, $L = 50 \mu\text{m}$ is the channel length, $H = 0.65 \text{ nm}$ is the channel height, and $W = 10 \mu\text{m}$ is the channel width. **New Figure 3g** suggests that the conductivity of monolayer MoS₂ after sulfur vacancies self-healing decreased sharply from 8.5×10^{-1} to $1.4 \times 10^{-3} \Omega^{-1}\text{cm}^{-1}$. As shown in new **Figure S** the mobility of electrons in the as-grown and self-healed MoS₂ is $\sim 0.96 \text{ cm}^2 \text{ V}^{-1} \text{ s}^{-1}$ and $\sim 0.26 \text{ cm}^2 \text{ V}^{-1} \text{ s}^{-1}$, respectively. Thus, electron concentrations varies about 643 times from 5.56×10^{19} to $8.65 \times 10^{16} \text{ cm}^{-2}$. In addition, due to the decrease of electron concentration, the threshold voltage dramatically shifted towards zero after the PSS-induced sulfur vacancy self-healing, as shown in **new Figure 3h**.

New figure 3g and S9. Output and transfer characteristics of a monolayer MoS₂ transistor, both before and after PSS-induced sulfur vacancy self-healing.

New Figure 3h. Transfer characteristics on the logarithmic scale of a monolayer MoS₂ transistor, both before and after PSS-induced sulfur vacancy self-healing.

6. The authors should clarify more Why there is a bigger gap between the $V_g=10V$ and $0V$, compared to other gating voltage curves in the $I_{ds}-V_{ds}$ plots in Figure 4b.

A: We thank the reviewer's comment. We consider that the key factor of the bigger gap is that in the measurement process, the pause time between each curve is not consistent. Since there are many defects on the surface of the SiO₂/Si substrate, which leads to the hysteresis effect [ACS Nano, 2012, 6 (6), 5635–5641].

New Figure S10a-c. **a**, Output characteristics of the homojunction at various V_G levels between 20 and -20V, along steps of 5V. **b**, Transfer characteristic of the homojunction at $V_D=1\text{V}$. **c**, Dependence on gate voltage of the drain current in dark and under different incident light intensity, at $V_D=1\text{V}$.

7. Why the source-drain current becomes hard to be tuned by negative gating as well as positive voltages as shown in the Figure 6a?

A: We thank the reviewer's comment. We are very sorry that we can't get the reviewer's meaning. However, according to the literal meaning of this sentence, we made two replies of

possible meanings. Firstly, the steady state of the source-drain current in the negative gating of the MoS₂ transistor is mainly due to light illumination. Light illumination can give MoS₂ films electron-hole pairs. In addition, at certain light intensity, gate voltage is difficult to change the numbers of electron-hole pairs generated by light illumination in **Figure R4**.

Figure R4. Gating response (I_{ds} - V_g) of the MoS₂ photodetector in dark and illuminated states, acquired for a backgate voltage V_g between -70 V and +40 V. Illumination power is 0.15 mW. Reproduced from reference [Nat. Nanotechnol. 8, 497-501 (2013)].

New figure 3g-h. Output and transfer characteristics of a monolayer MoS₂ transistor, both before and after PSS-induced sulfur vacancy self-healing.

New Figure 3f. Band diagram of the monolayer MoS₂ homojunction obtained from UPS measurements.

Secondly, the sulfur vacancy self-healing did not change the as-grown MoS₂ from n-type to p-type. As shown in **new Figure S3g-h**, the current decrease can be observed in the output characteristic curve and the threshold voltage dramatically shifted towards zero after the sulfur vacancy self-healing. Besides, the self-healed MoS₂ still showed a n-type semiconductor characteristic shown in **new Figure 3e**. Thus, our MoS₂ homojunction is not a

p-n junction but an n⁺-n junction. On the other hand, the UPS measurements also verifies this conclusion shown in **new Figure 3d**. Then, the as-grown MoS₂ region acted as an n⁺ type, and the self-healed region acted as an n-type. An n⁺-n monolayer MoS₂ homojunction was formed at the junction. In a word, our monolayer MoS₂ homojunction is not a p-n junction, but a n⁺-n junction, so it does not show an ambipolar transistor, the transfer characteristics of the homojunction is still characterized by n-type. So whether it is in the negative gate voltage or the positive gate voltages, the source-drain current will not appear a turning point to increase or decrease.

8. What is the response behavior of the untreated MoS₂ junction with /without light illumination? After a discharge, what would happen to the current decay time in an untreated MoS₂ junction.

A: We thank the reviewer's comment. Following this comments, we construct a monolayer as-grown MoS₂ transistor. Similar to the pervious MoS₂ homojunction shown in **Figure 3d**, the response time of the as-grown (untreated) MoS₂ transistor is also long. However, as shown in **Figure R5b**, the response time of the MoS₂ transistor can be improved to about 0.8 s by applying a short positive gate pulse. Thus, this experimental data verifies that the poor response performance of the MoS₂ homojunction can be explained in terms of trap states present either in MoS₂ or at the interface between the MoS₂ and the underlying SiO₂ substrate [Nat. Nanotechnol. 8, 497-501 (2013)]. This phenomenon has also previously reported in MoS₂ phototransistors, hybrid graphene-quantum dot phototransistors and phototransistors based on amorphous oxide semiconductors. [Nat. Nanotechnol. 7, 363-368 (2012) and Nat. Mater. 11, 301-305 (2012).]

Figure R5. **a**, Temporal photocurrent response of the as-grown (untreated) MoS₂ transistor to incident light. **b**, Temporal photocurrent response of the as-grown (untreated) MoS₂ transistor to incident light as the device is subjected to reset gate voltage pulse.

Reviewers' comments:

Reviewer #1 (Remarks to the Author):

The authors incorporated substantial new data to better support the claims of the manuscript. In the current version of the manuscript, I believe that the claims related to the vacancy healing are reasonably well supported. I recommend to minor revisions related to these claims, and other revisions related to other claims.

1. The combined XPS and HAADF STEM provide useful evidence in support of the claim of vacancy healing. However, statistics on densities of defects observed in HAADF STEM need to be provided to make a convincing statistical argument.

2. The shift in work function with PSS treatment, as measured by surface potential mapping, is consistent with other data in the paper and the overall interpretation. However, comparison of the scans in supplementary figure R2 does raise a concern. Specifically, the surface potential of the SiO₂ with respect to the metal contacts switches sign. This indicates that the surface charge on the oxide is different between the samples. The oxide surface charge also affects the doping. The authors should comment on how the processing steps and surface treatments influence the observed work functions. Regardless, the images do show a difference between PEDOT and PEDOT:PSS.

3. Page 9, line 229: "It is worth mentioning that our MoS₂ homojunction of ~150 meV barrier height has breakdown voltage up to -12 V (Supplementary Fig.11a), owing to the small barrier of the MoS₂ homojunction." The quoted breakdown voltage is meaningless without accounting for the series resistance noted above. It is important to identify the voltage across the junction at breakdown, not the voltage between the two contacts. The diode analysis should be redone by explicitly accounting for the series resistances of the contacts in the channel R_{series} by replacing V_D in equation 2 with $(V_D - I * R_{series})$. The authors should then redo the analysis of the ideality factor.

4. The response time of the photodetector is not particularly fast, despite the author's claims: Page 10, line 326: "The response speed is much faster than the MoS₂ based photoconductive photodetectors [39]..." The reference from 2013 is outdated. Faster response times have been reported in the literature since 2013, and the author should provide up-to-date references.

Reviewer #2 (Remarks to the Author):

I have reviewed the response letter and revised manuscript. The 5 comments raised on the original manuscript were well answered by the authors with additional material characterization and analysis on workfunction change. I recommend acceptance of the paper.

Reviewer #3 (Remarks to the Author):

I am satisfied with the amendment and recommend to accept it.

Reviewer #1 (Remarks to the Author):

The authors incorporated substantial new data to better support the claims of the manuscript. In the current version of the manuscript, I believe that the claims related to the vacancy healing are reasonably well supported. I recommend to minor revisions related to these claims, and other revisions related to other claims.

1. The combined XPS and HAADF STEM provide useful evidence in support of the claim of vacancy healing. However, statistics on densities of defects observed in HAADF STEM need to be provided to make a convincing statistical argument.

A: We thank the reviewer for raising this question. It's a good point. Following this comment, we calculate the sulfur vacancies concentration change through analyzing the XPS data. According to previous works [*Science* 350, 1065-1068 (2015), and *Nature materials* 15, 364 (2016)], XPS is a powerful tool to analyze defects concentration. The S/Mo ratios were determined from the integrated areas of the S 2p and Mo 3d peaks factored by their corresponding relative sensitivity factors. We measured the XPS peak area ratio of Mo 3d to S 2p states for the as-grown and self-healed MoS₂ shown in Fig. 2g and New Supplementary Fig. 8. The value of S:Mo ratio was increased from ~1.67 to ~1.86 by PEDOT:PSS solution treatment. That means that PSS-induced sulfur vacancies self-healing can significantly reduce the concentration of sulfur vacancies in monolayer MoS₂. Although the HAADF STEM technique is resultful in providing comprehensive information of monolayer MoS₂ defects at the atomic scale, the localized characterization of STEM is limited to present the real defects concentration in the entire materials.

Figure 2g. High-resolution XPS for Mo 3d before (top) and after (bottom) PSS treatment of MoS₂. Red and blue lines represent the intrinsic MoS₂ (i-MoS₂) and defective MoS₂ (d-MoS₂), respectively.

New Supplementary Figure 8. High-resolution XPS for S 2p before (top) and after (bottom) PSS treatment of MoS₂.

2. The shift in work function with PSS treatment, as measured by surface potential mapping, is consistent with other data in the paper and the overall interpretation. However, comparison of the scans in supplementary figure R2 does raise a concern. Specifically, the surface potential of the SiO₂ with respect to the metal contacts switches sign. This indicates that the surface charge on the oxide is different between the samples. The oxide surface charge also affects the doping. The authors should comment on how the processing steps and surface treatments influence the observed work functions. Regardless, the images do show a difference between PEDOT and PEDOT:PSS.

A: We thank the reviewer's suggestion. We believe that this phenomenon is mainly due to the accumulation of static electricity on the SiO₂ surface. On the other hand, surface potential measurement is extremely susceptible to electrostatic charge [*Nature Materials* 13, 1128–1134 (2014)]. When the electrostatic charge of the SiO₂ surface is removed by immersing in DI water, the surface potential of the re-measurement of the SiO₂ surface are returned to normal level shown in new Figure 1c.

Figure 1c and new Figure 1c. Previous (left) and now (right) corresponding 2D surface potential image of the device A₁.

3. Page 9, line 229: "It is worth mentioning that our MoS₂ homojunction of ~150 meV barrier height has breakdown voltage up to -12 V (Supplementary Fig.11a), owing to the small barrier of the MoS₂ homojunction." The quoted breakdown voltage is meaningless without accounting for the series resistance noted above. It is important to identify the voltage across the junction at breakdown, not the voltage between the two contacts. The diode analysis should be redone by explicitly accounting for the series resistances of the contacts in the channel R_{series} by replacing V_D in equation 2 with $(V_D - I * R_{series})$. The authors should then redo the analysis of the ideality factor.

A: We thank the reviewer's comment. Following the comment, we carried out several measurements and newly analyzed the experimental results. As shown in Figure R1, the curve at the linear scale suggests our device likely has a high breakdown voltage of -12V. In fact, at the logarithmic scale, the breakdown voltage of the device is much less than -12 V (the blue line in figure R1). Thus, the related sentence and the figure (Supplementary Fig. 11a) were removed. Although the contact resistance between 2D materials and electrodes is a great challenge, many researchers have published their investigation to reduce the series resistance. In our manuscript, the barrier height of the homojunction is our main claim which has been checked through various methods, such as UPS, SKPFM, and IV test at different temperature. All the above method can provide strong evidences of the junction ignoring the series resistance. Surely, the contact resistance of the two electrodes will influence the rectifying behavior of the device. As a result, we think that the contact resistance will not play a leading role and the breakdown voltage measurement is not an indispensable data for this manuscript. Finally, we must appreciate the reviewer for helping us to recognize this point.

Figure R1. Output characteristic on linear/logarithmic scale (black/blue) of the monolayer MoS₂ homojunction.

4. The response time of the photodetector is not particularly fast, despite the author's claims: Page 10, line 326: "The response speed is much faster than the MoS₂ based photoconductive

photodetectors [39]..." The reference from 2013 is outdated. Faster response times have been reported in the literature since 2013, and the author should provide up-to-date references.

A: We thank the reviewer for raising this question. Two latest papers [*Nat. Mater.* 16, 170-181 (2017) and *Adv. Funct. Mater.*, DOI: 10.1002/adfm.201603886 (2016)] have been added to support the claims.

Reviewer #2 (Remarks to the Author):

I have reviewed the response letter and revised manuscript. The 5 comments raised on the original manuscript were well answered by the authors with additional material characterization and analysis on work function change. I recommend acceptance of the paper.

A: We are very grateful to the reviewers for their recognition of our work.

Reviewer #3 (Remarks to the Author):

I am satisfied with the amendment and recommend to accept it.

A: We are very grateful to the reviewers for their recognition of our work.

REVIEWERS' COMMENTS:

Reviewer #1 (Remarks to the Author):

The authors have sufficiently addressed my concerns.

Reviewer #1 (Remarks to the Author):

The authors have sufficiently addressed my concerns.

A: We are very grateful to the reviewers for their recognition of our work.